# Efficient identification of informative features in simulation-based inference

**Jonas Beck**
University of Tübingen
jonas.beck@uni-tuebingen.de

**Michael Deistler**
University of Tübingen
michael.deistler@uni-tuebingen.de

**Yves Bernaerts**
University of Tübingen
yves.bernaerts@uni-tuebingen.de

**Jakob H. Macke**
University of Tübingen
Max Planck Institute for Intelligent Systems
jakob.macke@uni-tuebingen.de

**Philipp Berens**
University of Tübingen
philipp.berens@uni-tuebingen.de

## Abstract

Simulation-based Bayesian inference (SBI) can be used to estimate the parameters of complex mechanistic models given observed model outputs without requiring access to explicit likelihood evaluations. A prime example for the application of SBI in neuroscience involves estimating the parameters governing the response dynamics of Hodgkin-Huxley (HH) models from electrophysiological measurements, by inferring a posterior over the parameters that is consistent with a set of observations. To this end, many SBI methods employ a set of summary statistics or scientifically interpretable features to estimate a surrogate likelihood or posterior. However, currently, there is no way to identify how much each summary statistic or feature contributes to reducing posterior uncertainty. To address this challenge, one could simply compare the posteriors with and without a given feature included in the inference process. However, for large or nested feature sets, this would necessitate repeatedly estimating the posterior, which is computationally expensive or even prohibitive. Here, we provide a more efficient approach based on the SBI method neural likelihood estimation (NLE): We show that one can marginalize the trained surrogate likelihood post-hoc before inferring the posterior to assess the contribution of a feature. We demonstrate the usefulness of our method by identifying the most important features for inferring parameters of an example HH neuron model. Beyond neuroscience, our method is generally applicable to SBI workflows that rely on data features for inference used in other scientific fields.

## 1 Introduction

Mechanistic models are an elegant way to encode scientific knowledge about the world in the form of numerical simulations. They include models such as Kepler's laws of planetary motion [1], the SEIR model [2] for describing the spread of infectious diseases or the Hodgkin-Huxley (HH) model for the dynamics of action potentials in neurons [3]. Efficiently constraining the parameters of such models by measurements is a key problem in many disciplines [4–7]. Since these models give rise to intractable likelihood functions, however, classical likelihood-based Bayesian methods such as variational inference [8] or Markov Chain Monte Carlo [9] cannot be used directly. There are algorithms,

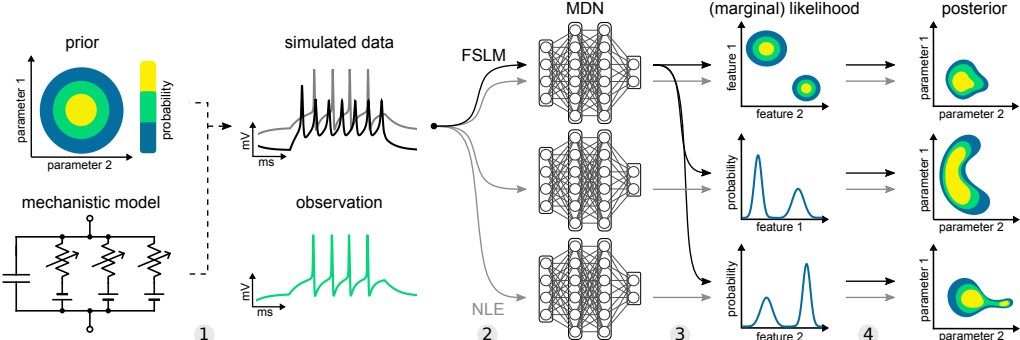

Figure 1: Feature Selection Through Likelihood Marginalization (FSLM) is a method to identify informative features in simulation-based inference (SBI). It builds on Neural Likelihood Estimation (NLE) with mixture density networks (MDN). This requires a prior over the parameter space, a mechanistic model to simulate data and an observed data point. **1.** Parameters are sampled from the prior and used to simulate a synthetic dataset. **2.** A MDN learns the probabilistic relationship between data (or data features) and underlying parameters, in the form of a tractable surrogate likelihood. **3.** After the MDN has learned to approximate the likelihood, it can be conditioned on the observation to yield a likelihood estimate that is parameterised as a mixture of Gaussians. **4.** This surrogate likelihood is then combined with the prior distribution to obtain the posterior distribution, i.e. the space of parameters consistent with both prior knowledge and data. Consistent parameters are assigned a high, inconsistent parameters a low probability. The naive approach for identifying the importance of features would be to repeat **2.** for different sets of data features (grey). With FSLM, the likelihood estimate can be marginalized post-hoc and a single estimate is therefore sufficient to obtain multiple posterior distributions (black).

such as ABC-MCMC [10] or pseudo-marginal methods [11] that deal with this problem, however, they have potentially slow convergence rates and are computationally expensive. To overcome this problem, several techniques collectively known as simulation-based inference (SBI) have recently been developed [12]. Leveraging the ability to simulate the model, these techniques obtain estimates of the likelihood or posterior from simulated data. The most recent of these algorithms such as neural likelihood estimation (NLE) [13] or neural posterior estimation (NPE) [14–16] employ state-of-the-art neural density estimators to learn tractable surrogates of these functions to be evaluated instead of the real quantities. To this end, summary statistics or features capturing essential aspects of the model's dynamics are defined by the scientist to reduce the high dimensional model output to manageable scale, in turn decreasing the problem complexity and computation costs. This is often done by hand to emphasize specific aspects of the data or to aid scientific interpretation [12], but can also be automated [17–23].

In neuroscience, many different approaches have been developed in order to find suitable parameters of models of neural activity [4, 5, 24–29]. SBI approaches have been used to infer parameters in biophysical neuron models from measurements of neural activity in a Bayesian way [15, 30, 31]. For inference of HH models from electrophysiological data, features such as action potential threshold or width and resting membrane potential or spike count can be used, reflecting measures that electrophysiologists use to characterize recorded neurons.

For scientific interpretation of SBI results, it is often of interest which features, or combinations of features, have the biggest impact on the posterior and which parameters they affect specifically. To this end, one could compare the posterior uncertainty estimated with and without including a specific feature in the SBI method — the increase in uncertainty resulting from not relying on that feature can be used as a measure of its importance and on average is equivalent to its mutual information. Of course, this approach can also be applied to whole subsets of features. To evaluate an entire set of features exhaustively would require re-estimating the posterior many times (Fig. 1, grey), which would scale prohibitively with the number of features. To address this issue, we here introduce a method called Feature Selection Through Likelihood Marginalization (FSLM) to compute posteriors for arbitrary subsets of features, without the need for repeated training (Fig. 1, black). To achieve this, we here use NLE [32] and exploit the marignalization properties of mixture density networks

(MDNs) [33], which can be used with NLE as a density estimator: instead of re-estimating the surrogate likelihood from scratch, we marginalize it analytically with respect to a given (set of) features before applying Bayes rule to obtain the posterior estimate. This way, we can efficiently compare the posterior uncertainty with and without including a certain feature.

For a simple linear Gaussian model and non-linear HH models, we show that the obtained posterior estimates are as accurate and robust as when repeatedly retraining density estimators from scratch, despite being much faster to compute. We then apply our algorithm to study which features are the most useful for constraining posteriors over parameters of the HH model. Our tool allows us also to study which model parameter is affected by which summary feature. Finally, we suggest a greedy feature selection strategy to select useful features from a predefined set.

## 2 Methods

### 2.1 Neural Likelihood Estimation (NLE)

Neural Likelihood Estimation [32] is a SBI method which approximates the likelihood $p(\boldsymbol{x} \mid \boldsymbol{\theta})$ of the data $\boldsymbol{x}$ given the model parameters $\boldsymbol{\theta}$ by training a conditional neural density estimator on data generated from simulations of a mechanistic model. Using Bayes rule, the approximate likelihood can then be used to obtain an estimate of the posterior (Fig. 1). Unlike NPE [15, 16], NLE requires an additional sampling or inference step [34] to obtain the posterior distribution. However, it allows access to the intermediate likelihood approximation, a property we will later exploit to develop our efficient feature selection algorithm (see Sec. 2.3).

First, a set of N parameters are sampled from the prior distribution $\boldsymbol{\theta}_n \sim p(\boldsymbol{\theta})$, $n \in \{1, \ldots, N\}$. With these parameters, the simulator is run to implicitly sample from the model's likelihood function according to $\boldsymbol{x}_n \sim p(\boldsymbol{x}|\boldsymbol{\theta}_n)$. Here, $\boldsymbol{x}_n = (x_1, \ldots, x_{N_f})$ are feature vectors that are usually taken to be a function of the simulator output $\boldsymbol{s}$ with the individual features $x_i = f_i(\boldsymbol{s})$, where $i \in \{1, \ldots, N_f\}$, rather than the output directly. For mechanistic models whose output are time series, $f$ also reduces the dimensionality of the data to a set of lower dimensional data features. The resulting training data $\{\boldsymbol{x}_n, \boldsymbol{\theta}_n\}_{1:N} \sim p(\boldsymbol{x}, \boldsymbol{\theta})$ can then be used to train a conditional density estimator $q_\phi(\boldsymbol{x} \mid \boldsymbol{\theta})$, parameterized by $\phi$, to approximate the likelihood function. Here, we use a Mixture Density Network for this task [33], the output density of which is parameterised by a mixture of Gaussians, which can be marginalized analytically. Thus, $\hat{p}(\boldsymbol{x} \mid \boldsymbol{\theta}) = \sum_k \pi_k \mathcal{N}(\boldsymbol{\mu_k}, \boldsymbol{\Sigma_k})$ with the parameters $(\boldsymbol{\mu_k}, \boldsymbol{\Sigma_k}, \boldsymbol{\pi})$ being non-linear functions of the inputs $\theta$ and the network parameters $\phi$. The parameters $\phi$ are optimized by maximizing the log-likelihood $\mathcal{L}(\boldsymbol{x}, \boldsymbol{\theta}) = \frac{1}{N} \sum_n \log q_\phi(\boldsymbol{x} \mid \boldsymbol{\theta})$ over the training data with respect to $\phi$. As the number of training samples goes to infinity, this is equivalent to maximizing the negative Kullback-Leibler (KL) divergence between the true and approximate posterior for every $\boldsymbol{x} \in \text{supp}(\boldsymbol{x})$ [32]:

$$\mathbb{E}_{p(\boldsymbol{\theta}, \boldsymbol{x})} \left[\log q_\phi(\boldsymbol{x}|\boldsymbol{\theta})\right] = -\mathbb{E}_{p(\boldsymbol{\theta})} \left[\mathcal{D}_{KL}(p(\boldsymbol{x}|\boldsymbol{\theta})|| q_\phi(\boldsymbol{x}|\boldsymbol{\theta}))\right] + const. \tag{1}$$

After obtaining such a tractable likelihood surrogate, Bayes rule can be used to obtain an estimate of the posterior conditioned on an observation $\boldsymbol{x}_o$ (see Eq. 2) for instance via Markov Chain Monte Carlo (MCMC):

$$\hat{p}(\boldsymbol{\theta} \mid \boldsymbol{x}_o) \propto q_\phi(\boldsymbol{x}_o \mid \boldsymbol{\theta})p(\boldsymbol{\theta}) \tag{2}$$

### 2.2 A naive algorithm for quantifying feature importance

Given this posterior estimate, we would now like to answer the following question: for which feature $x_i$ from a vector of features $\boldsymbol{x}$ does the uncertainty of the posterior estimate $\hat{p}(\boldsymbol{\theta} \mid \boldsymbol{x})$ increase the most, when it is ignored? For a single feature, a naive and costly algorithm does the following: iterate over $\boldsymbol{x}$ to obtain $\boldsymbol{x}_{\backslash i} = (x_1, \ldots x_{i-1}, x_{i+1}, \ldots, x_N)$, train a total of N+1 density estimators to obtain the likelihoods $\hat{p}(\boldsymbol{x}_{\backslash i}|\theta)$ and $\hat{p}(\boldsymbol{x}|\theta)$, sample their associated posteriors and compare $\hat{p}(\theta|\boldsymbol{x}_{\backslash i})$ to the reference posterior $\hat{p}(\boldsymbol{\theta} \mid \boldsymbol{x})$ with every feature present. The same procedure can also be applied to quantify the contribution of any arbitrary subset of features. As this procedure requires estimating the likelihood based on the reduced feature set from scratch for each feature (set) (Fig. 1, grey arrows), it is computationally costly. To quantify the contribution of different features more efficiently, we would thus need a way to avoid re-estimating the posterior with each feature left out.

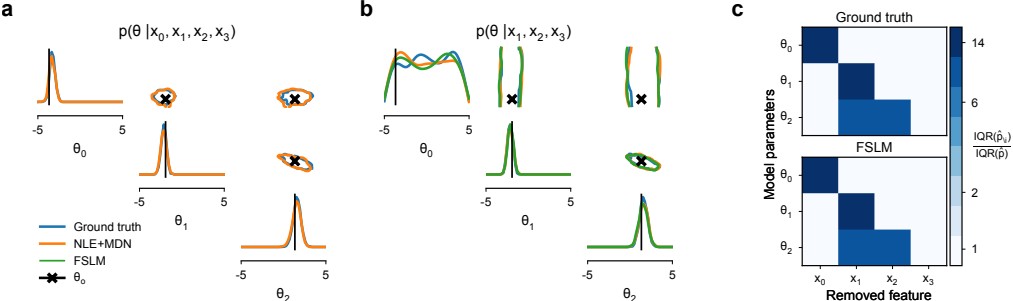

Figure 2: FSLM can accurately compute posteriors with one feature marginalized out for a linear model described in Eq. 3: **a.** 1D and 2D marginals of the full posterior estimated by NLE and the ground truth, which can be analytically computed. **b.** Posterior distribution when $x_0$ is removed from the feature set. FSLM is as accurate as re-estimating NLE on the reduced feature set. **c.** Increase in the uncertainty of the marginal posterior distributions measured using the IQR ratio (as defined in Sec. 2.4) for each feature, for the analytical posteriors and those computed using FSLM. The joint influence of $x_1$ and $x_2$ on $\theta_2$, as per our model, is clearly visible.

## 2.3 Efficient quantification of feature importance through post-hoc likelihood marginalization

To more efficiently estimate posteriors for any subset of features, we propose to marginalize the likelihood estimate obtained via NLE post-hoc, as opposed to training a separate density estimator for every new set of features to obtain the marginal estimates (Sec. 2.2, Fig. 1). We refer to this algorithm as efficient Feature Selection through Likelihood Marginalization (FSLM).

Suppose we want to evaluate the joint increase in posterior uncertainty for a subset of features. We partition our feature vector $x = [x_1, x_2]$, such that $x_2$ contains the features to be removed from the inference. To avoid the need for training the surrogate likelihood $q_\phi$ for such a feature subset, we can rewrite the posterior with respect to $x_1$ as

$$\hat{p}(\boldsymbol{\theta} \mid \boldsymbol{x}_1) \propto q_\phi(\boldsymbol{x}_1 \mid \boldsymbol{\theta}) p(\boldsymbol{\theta}) = \int q_\phi(\boldsymbol{x}_1, \boldsymbol{x}_2 \mid \boldsymbol{\theta}) d\boldsymbol{x}_2 \, p(\boldsymbol{\theta}) \tag{3}$$

Since we parameterize the approximate likelihood $q_\phi$ as an MDN, we can perform this marginalization analytically: For a K-component MDN, the parameters of each mixture component can be similarly partitioned to $x$ [35, Chapter 2.3.2]:

$$\boldsymbol{\mu}_k = [\boldsymbol{\mu}_{k,1}, \boldsymbol{\mu}_{k,2}], \quad \boldsymbol{\Sigma}_k = \begin{bmatrix} \boldsymbol{\Sigma}_{k,11} & \boldsymbol{\Sigma}_{k,12} \\ \boldsymbol{\Sigma}_{k,21} & \boldsymbol{\Sigma}_{k,22} \end{bmatrix}, \quad k \in \{1, ..., K\}. \tag{4}$$

Then the marginalization with respect to $x_2$ results in the following posterior distribution

$$\hat{p}(\boldsymbol{\theta} \mid \boldsymbol{x}_1) \propto q_\phi(\boldsymbol{x}_1 \mid \boldsymbol{\theta}) p(\boldsymbol{\theta}) = \sum_{k=1}^{K} \pi_k \mathcal{N}_k(\boldsymbol{x}_1 \mid \boldsymbol{\mu}_{k,1}, \boldsymbol{\Sigma}_{k,11}) \, p(\boldsymbol{\theta}), \tag{5}$$

where $\boldsymbol{\pi} = \boldsymbol{\pi}(\boldsymbol{\theta})$, $\boldsymbol{\mu} = \boldsymbol{\mu}(\boldsymbol{\theta})$ and $\boldsymbol{\Sigma} = \boldsymbol{\Sigma}(\boldsymbol{\theta})$. FSLM can thus sample from the posterior given arbitrary feature subsets without estimating the surrogate likelihood $q_\phi$ from scratch.

We implement FSLM using python 3.8, building on top of the public sbi library [36]. All the code is available at github.com/berenslab/fslm_repo. All computations were done on an internal cluster running Intel(R) Xeon(R) Gold 6226R CPUs @ 2.90GHz.

## 2.4 Measures of posterior uncertainty

Assuming the NLE procedure converges and the density estimator is sufficiently flexible, differences in posterior uncertainty can be ascribed to differences in information content of features or noise in the data. Given that that noise in the data remains constant, relative differences in posterior uncertainty can be used to assess the contributions of individual summary features. Hence, we can identify informative summary statistics by quantifying the change in uncertainty of $\hat{p}(\boldsymbol{\theta}|\boldsymbol{x}_{\setminus i})$ with respect to $\hat{p}(\boldsymbol{\theta} \mid \boldsymbol{x})$. For this we use two metrics:

| Model | Method | Training time | Sampling time | Total time | KL |
|---|---|---|---|---|---|
| LGM [min] | NLE + MDN | $3.57 \pm 0.79$ | $\mathbf{0.33 \pm 0.04}$ | $3.89 \pm 0.80$ | $0.06 \pm 0.10$ |
| | NLE + MAF | $11.40 \pm 2.22$ | $0.57 \pm 0.28$ | $11.98 \pm 2.30$ | $\mathbf{0.05 \pm 0.12}$ |
| | FSLM | $\mathbf{0.64 \pm 0.21}$ | $0.66 \pm 0.10$ | $\mathbf{1.30 \pm 0.26}$ | $0.07 \pm 0.16$ |
| HH [h] | NLE + MDN | $84.20 \pm 9.28$ | $\mathbf{10.34 \pm 0.54}$ | $94.53 \pm 9.24$ | $29.04 \pm 8.13$ |
| | NLE + MAF | $130.54 \pm 29.18$ | $19.58 \pm 4.64$ | $150.12 \pm 30.86$ | $\mathbf{27.96 \pm 10.15}$ |
| | FSLM | $\mathbf{7.21 \pm 1.49}$ | $13.27 \pm 1.04$ | $\mathbf{20.48 \pm 2.08}$ | $28.27 \pm 7.40$ |

Table 1: Performance of FSLM (mean ± standard deviation across 10 runs), compared to repeated runs of NLE employing a MDN or MAF. For the LGM example, all times are minutes [min] and rejection sampling was used to sample the posterior. For the HH model, all times are hours [h] and MCMC was used. The last column (KL) measures the KL divergence (as defined in Sec. 2.4). For the LGM KL is measured between the posterior and the analytic ground truth. Since no ground truth data is available for the HH model, we use a NLE+MDN estimate of the posterior trained on a substantially larger set of summary statistics (see Appendix, Tab. 3). The mean and standard deviation were obtained across all 10 runs and for all subsets that exclude one feature. The timings are single threaded.

**Ratio of IQR of the marginals:** We use the ratio of inter quantile ranges (IQR) of the 1D-marginals of $\hat{p}(\boldsymbol{\theta} \mid \boldsymbol{x}_1)$ and $\hat{p}(\boldsymbol{\theta} \mid \boldsymbol{x})$. This lets us deduce if specific features $x_i$ reduce the uncertainty of some parameters $\theta_j$ more than others. We use IQRs as a measure of spread as it is much less susceptible to outliers compared to the standard deviation. The resulting visualization allows to precisely pin down which parameter is affected by which summary feature.

**Non-parametric estimate of KL divergence:** As a second metric, we use a non-parametric estimate of the KL divergence between $\hat{p}(\boldsymbol{\theta} \mid \boldsymbol{x}_1)$ and $\hat{p}(\boldsymbol{\theta} \mid \boldsymbol{x})$. Since a posterior estimate, on average, cannot become more constrained if features are removed, increases in the KL estimate indicate an increase in uncertainty overall. To estimate the KL-divergence between $\hat{p}(\boldsymbol{\theta} \mid \boldsymbol{x}_1)$ and $\hat{p}(\boldsymbol{\theta} \mid \boldsymbol{x})$, we employ a purely sample based estimator [37, 38]. This is necessary because NLE only allows to evaluate unnormalized posteriors which circumvents having to evaluate them explicitly. The estimator makes use of 1-nearest neighbor search and is implemented using k-d trees [39], to reduce the $2^N$ comparison operations to a manageable number. With samples $\boldsymbol{X} = \{X_i\}_{i=1}^N$ and $\boldsymbol{Y} = \{Y_i\}_{i=1}^M$ from $\hat{p}(\boldsymbol{\theta} \mid \boldsymbol{x}_1)$ and $\hat{p}(\boldsymbol{\theta} \mid \boldsymbol{x})$ respectively, the KL-divergence $KL(\hat{p}(\boldsymbol{\theta} \mid \boldsymbol{x}_1)\|\hat{p}(\boldsymbol{\theta} \mid \boldsymbol{x}))$ is estimated as

$$\mathcal{D}_{KL} = \frac{d}{N} \sum_{i=1}^N \log \frac{\min_j ||X_i - Y_j||}{\min_{j \neq i} j ||X_i - X_j||} + \log \frac{M}{N-1}, \tag{6}$$

where $d$ is the dimensionality of the samples, and $|| \cdot ||$ denotes the $\ell_2$-norm. We note that the Maximum Mean Discrepancy (MMD) [40, 41] could be used as an alternative measure, but preliminary experiments indicated that the KL estimate was more consistent.

## 3 Results

### 3.1 Example 1: Linear Gaussian model

We illustrate our algorithm in a simple toy problem, where by construction, we know which features should have which effect. We used a Linear Gaussian Model (LGM) with a three dimensional parameter space $\boldsymbol{\theta} = [\theta_0, \theta_1, \theta_2]$ and four dimensional feature space $\boldsymbol{x} = [x_0, x_1, x_2, x_3]$. $\boldsymbol{\theta}$ is drawn from a uniform prior $\theta_i \sim \mathcal{U}(-5, 5)$, $i \in \{0, 1, 2\}$, with the features $\boldsymbol{x}$ being noisy linear transformations of the parameter vector $\boldsymbol{\theta}$. The mean $\boldsymbol{\mu}(\boldsymbol{\theta})$ and covariance $\boldsymbol{\Sigma}$ are defined by

$$\boldsymbol{\mu}(\boldsymbol{\theta}) = \boldsymbol{\mu}_0 + \boldsymbol{L}\boldsymbol{\theta}, \quad \boldsymbol{\Sigma} = \sigma^2 \mathbb{1}, \quad \boldsymbol{L} = \begin{bmatrix} 1 & 0 & 0 \\ 0 & 1 & 0 \\ 0 & 1 & 1 \\ 0 & 0 & 0 \end{bmatrix}. \tag{7}$$

With scale $\sigma$ and shift $\boldsymbol{\mu}_0$. We construct $\boldsymbol{L}$ such that we can easily verify that the posterior reacts to and interacts with marginalized features as expected. In our example, we let feature $x_0$ solely depend on $\theta_0$, $x_1$ on both $\theta_1$ and $\theta_2$, $x_2$ on $\theta_2$, and $x_3$ without influence on any model parameter.

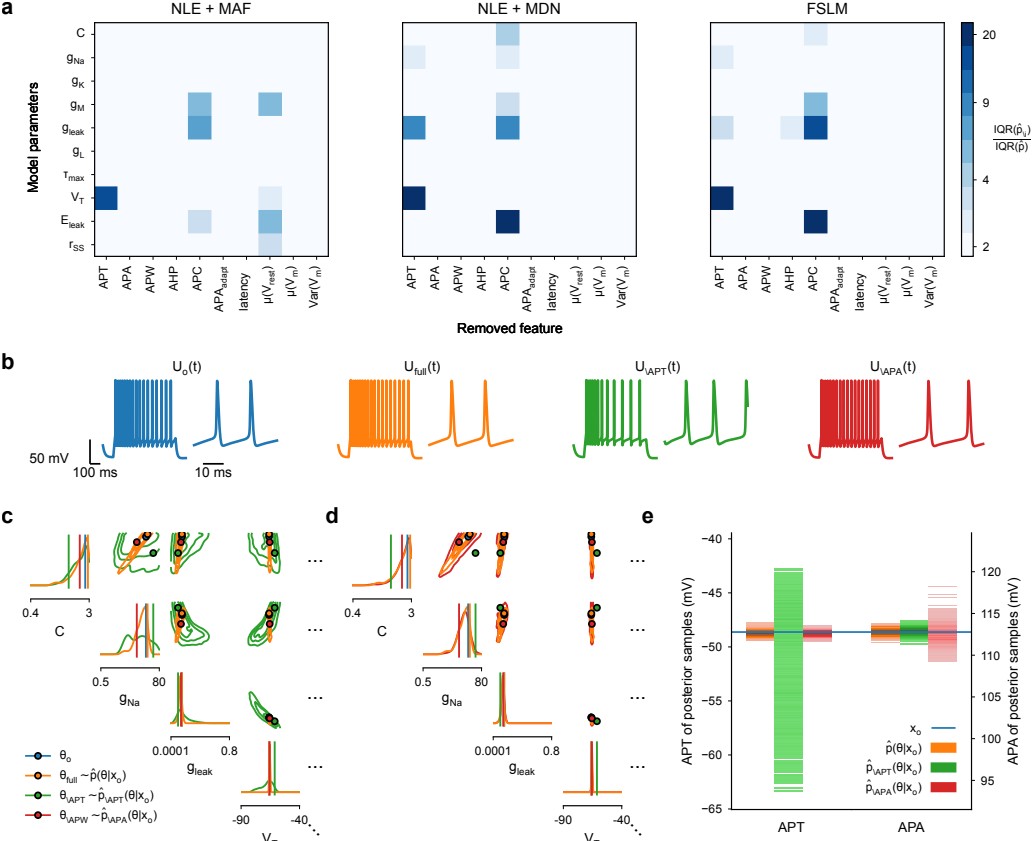

Figure 3: Removal of individual features affects posterior uncertainty and samples in a HH model: **a.** The ratio of IQRs for the marginal posterior distributions (as defined in Sec. 2.4), when individual features are left out. Values are large (dark) when the feature has a large effect on the marginal posterior for the specified model parameter, and small otherwise (light). **b.** Simulations from the HH model with the original true parameters ($\theta_o$, blue), parameters drawn from the full posterior ($\theta_{full}$, orange) and two posterior distributions when leaving out the features APT and APA ($\theta_{\backslash APT}$, green and red, $\theta_{\backslash APA}$, respectively). **c.** Posterior distribution with the full feature set (orange) and after removal of the highly constraining feature APT (green) shows the effect on some of the 1D and 2D marginals (see Fig.6 for complete posteriors) compared to the full posterior (orange). **d.** As in **c.** for the less informative feature APW. **e.** The effect of the removal of informative vs. uninformative features from the posterior estimate on the feature values of the simulated parameter samples.

The resulting posterior is well concentrated on $\theta_o$ and the NLE+MDN estimate matches the analytically computed posterior well (Fig. 2**a**). To show that FSLM can correctly identify individual feature importance, while being more efficient than re-running NLE for each subset, we compared samples from the FSLM-derived posterior estimates for every three feature subset to those obtained using the analytical marginal likelihood $p(\boldsymbol{x}_{\setminus i} \mid \boldsymbol{\theta})$, $i \in \{0, 1, 2, 3\}$ and brute-force NLE estimates using separate $q_\phi(\boldsymbol{x}_{\setminus i} | \boldsymbol{\theta})$ for each subset (Fig. 2**b**, Appendix Fig.5 **a-c**). The brute-force baseline was computed with both a MDN and a Masked Autoregressive Flow (MAF) [42]. The MDNs had three hidden layers and 10 mixture components. The MAF consisted of five 3 layer deep MADEs [43]. Training was done on 10,000 samples, then 500 samples were drawn from each posterior, for a total of 10 different initializations. Training was terminated when the log-likelihoods had stopped improving on a validation set that was split from the training set at 10% over 20 epochs.

The posterior estimates obtained from FSLM matched those from the analytic ground truth very accurately (Fig. 2**b-e**, Appendix Fig. 5**a-c** and Tab. 1), while being several times faster to compute than both NLE alternatives (see Tab. 1). As expected from the construction of our model, we observed that $\hat{p}(\boldsymbol{\theta} \mid \boldsymbol{x}_{\setminus 0})$ was not only less constrained compared to $p(\boldsymbol{\theta} \mid \boldsymbol{x})$ (Fig. 2**a**), but that $\hat{p}(\theta_0 \mid \boldsymbol{x}_{\setminus 0}) = p(\theta_0) = \mathcal{U}(-5, 5)$. We also found that FSLM correctly identified the dependency structure between the other parameters and features (Appendix Fig. 5**a-c**), with FSLM matching the analytic ground truth while being much faster than both NLE approaches (Tab. 1).

## 3.2 Example 2: Hodgkin-Huxley neuron model

Next, we turned to a more realistic complex mechanistic model used for simulations of neurons. For our experiments we used a single compartment Hodgkin-Huxley model [3], Eq. 8, derived from Pospischil et al. [44]. In addition to the standard sodium, potassium and leak currents to generate spiking this model also considers a slow non-inactivating $K^+$ current to model spike-frequency adaptation and a high-threshold $Ca^{2+}$ current to generate bursting.

$$
\begin{aligned}
I_t = C\, A_{Soma} \frac{dV_t}{dt} &+ \bar{g}_{Na} m^3 h (V_t - E_{Na}) + \bar{g}_K n^4 (V_t - E_K) \\
&+ \bar{g}_{leak}(V_t - E_{leak}) + \bar{g}_M p(V_t - E_K) + \bar{g}_L q^2 r (V_t - E_{Ca})
\end{aligned}
\tag{8}
$$

Eq. 8 models the evolution of the membrane voltage $V_t = V(t)$ given a stimulus $I(t) = I_t$. $\bar{g}_i$, $i \in \{Na, K, l, M, L\}$ are the maximum conductances of the sodium, potassium, leak, adaptive potassium and calcium ion channels, respectively, and $E_i$ are the associated reversal potentials. $I_t$ denotes the current per unit area, $C$ the membrane capacitance, $A_{Soma}$ the compartment area and $n, m, h, q, r$ and $p$ represent the fraction of independent gates in the open state, based on Hodgkin and Huxley [3]. For the remaining equations and parameter settings, see Appendix A.3.

We then compared the performance of FSLM and the two versions of brute-force NLE to measure how commonly used summary features constrain the posterior of an exemplary voltage trace. Specifically, we used a set of 10 summary statistics commonly of interest to neurophysiologists, which capture the response dynamics of the HH model and can be used to differentiate cortical cell types. They are briefly explained in Table 3. To extract these features from the simulated traces we use the Allen SDK.

To train the density estimator we simulated 1 million observations with parameters drawn from a Uniform prior over biologically plausible parameter ranges (see A.3). However, because a substantial fraction of the generated observations did not produce spiking behavior, some features such as spike rate were not defined. Hence, these observations could not be used to train the density estimator. We employed two different techniques introduced by Lueckmann et al. [15] to reduce the amount of invalid samples returned by the prior and to compensate for any remaining invalid observations in the training data (for details see Appendix A.2). This yielded accurate posterior estimates, despite incomplete data. The change in uncertainty was then assessed on 3000 samples from the MDN estimates of $\hat{p}(\boldsymbol{\theta} \mid \boldsymbol{x})$ and $\hat{p}(\boldsymbol{\theta} \mid \boldsymbol{x}_{\setminus i})$. Architectures and training were kept the same as in Sec.3.1

We found that all three algorithms identified the same two features as yielding the highest increase in posterior uncertainty (Fig. 3**a**): the action potential threshold (APT) and the action potential count (APC), while a MAF additionally identified the mean resting potential ($\mu(V_{rest})$) as a constraining feature. For the MDN-based density estimates, APT strongly affected the threshold voltage $V_T$, the

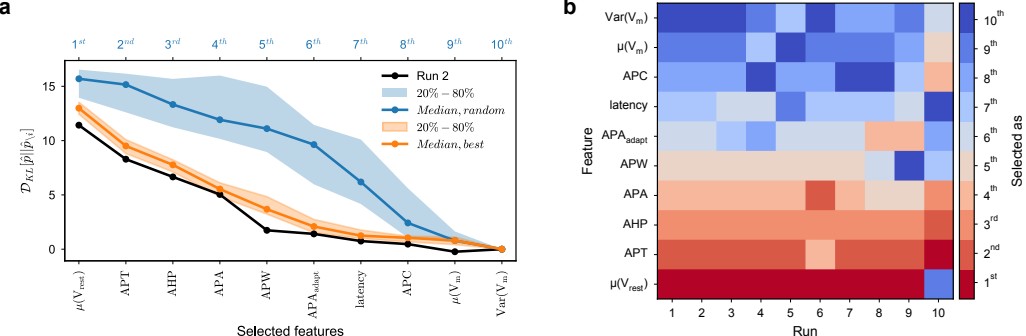

Figure 4: Greedy selection of the most informative features from a candidate set for the HH model: **a.** Example trajectory of a single run using FSLM for greedy search (black). Parameters are added to the feature set from left to right (right most point includes all features). Medians and confidence intervals for 50 runs, when parameters are selected at random (blue) vs. when selected by the greedy-search procedure (orange). **b.** Order in which the features were chosen for the first 10 feature searches.

leak conductance $g_{leak}$ as well as the sodium conductance $g_{Na}$. APC, influenced the leak reversal potential $E_{leak}$, the leak conductance $g_{leak}$, as well as $C$ and $g_{Na}$ (Fig. 3**a**). When a flow was used, $g_{NA}$ did not seem to be constrained by either APT or APC and $\mu(V_{rest})$ additionally constrained the leak reversal potential ($E_{leak}$), as well as the adaptive Potassium current $g_M$. In contrast, the action potential amplitude (APA), as well as other features, did not have a sizable impact on the posterior in this example (Fig. 3**a**). While the results for both MDN-based algorithms are equivalent and have strong overlap with the flow-based predictions, FSLM was significantly faster than both brute-force alternatives ($\approx$ 5-fold and $\approx$ 7-fold speedups respectively, Table 1) and overall almost as accurate, when measured against a common baseline (Table 1).

As a consequence, simulations based on samples from $\hat{p}(\boldsymbol{\theta} \mid \boldsymbol{x})$ and $\hat{p}(\boldsymbol{\theta} \mid \boldsymbol{x}_{\setminus APA})$ were comparable to the observed voltage trace $V_o(t)$, while the sample drawn from $\hat{p}(\boldsymbol{\theta} \mid \boldsymbol{x}_{\setminus APT})$ showed different spiking behavior, with much longer inter spike intervals in the simulation (Fig. 3**b**). This resulted from higher values for the sodium conductance $g_{Na}$ and threshold voltage $V_T$, compared to $\theta_o$, for the particular simulation shown (Fig. 3**a,c**). The measured increase in uncertainty is also directly visible in the two-dimensional marginal distributions of the posterior, where the removal of APT lead to a much less constrained posterior estimate (Fig.3**c**), while the removal of APA did not affect the posterior uncertainty much (Fig.3**d**). Also, the measured summary statistics from samples from $\hat{p}(\boldsymbol{\theta} \mid \boldsymbol{x}_{\setminus APT})$ and $\hat{p}(\boldsymbol{\theta} \mid \boldsymbol{x}_{\setminus APA})$ showed the expected behavior: the APT feature was much more variable in samples from $\hat{p}(\boldsymbol{\theta} \mid \boldsymbol{x}_{\setminus APT})$ than in $\hat{p}(\boldsymbol{\theta} \mid \boldsymbol{x})$, while the estimated value for APA did not vary as drastically even for samples from $\hat{p}(\boldsymbol{\theta} \mid \boldsymbol{x}_{\setminus APA})$ (Fig. 3**e**).

## 3.3 Efficient greedy feature selection using FSLM

In the previous section, we showed that FSLM can be used to study how the removal of individual features increases the uncertainty of the posterior. Next we show how FSLM can be used to study how the addition of features decreases uncertainty and provide a method to efficiently search for informative features by greedy selection of new features.

First, we used NLE to obtain a posterior estimate given a candidate set of features. We then used FSLM to obtain all the 1D marginal likelihoods, sampled their associated posteriors $\hat{p}(\boldsymbol{\theta} \mid x_i)$ and computed their KL divergences to the full posterior. We subsequently selected the feature $x_i$ that minimizes the KL, before we repeated the procedure for all 2D marginals $\hat{p}(\boldsymbol{\theta} \mid (x_i, x_j))$ which include the selected feature $x_i$. We can iterate this procedure until a desired number of features is identified. This is equivalent to a beam-search [45] with a beam size equal to one. To investigate combinations of features the scheme is easily extensible to larger beam sizes.

To demonstrate this, we turned again to the HH example already studied in Sec. 3.2. We used 1 million samples for training and sampled every posterior 1000 times. To check for consistency, we repeated the procedure 50 times. Compared to selecting features at random, selecting features based

on the results of our greedy algorithm consistently yielded more constrained posterior estimates (see Fig. 4**a**). Furthermore, we found the three most informative features to be $\mu(V_{rest})$, $APT$, $AHP$, respectively, which were consistently chosen in this order (see Fig. 4**b**).

## 4  Discussion

We set out to develop an efficient method to characterize the relative contribution of different features to constraining the posterior in simulation-based inference. To this end, we used NLE with a mixture density network to estimate the surrogate likelihood and marginalize it analytically. We showed that our method, FSLM, correctly identifies influential features both in a simple linear Gaussian model, as well as in a complex mechanistic model, namely the HH model in neuroscience. In both cases FSLM was as accurate as repeated application of NLE, but many times faster, independent of whether the likelihood was approximated by MDNs or MAFs. While the different architectures lead to subtle differences in their predictions of important features of HH models, we found them still to be mostly in agreement. FSLM can thus provide guidance on how to specify, prepare and select appropriate summary statistics, while also allowing to study which parameters are constrained by which features to gain insight into the scientific structure of the model. In contrast to methods developed for automatically deriving optimal (combinations of) summary features [17–19, 21, 23], FSLM focuses on making the evaluation and selection of individual scientifically relevant feature sets interpretable. Our visualization method allowed us to identify which features influenced the posterior uncertainty over which model parameters, and for which model parameters the posterior was not influenced by any feature. A similar approach could also be implemented for Approximate Bayesian Computation (ABC) [46, 47].

Compared to approaches that aim to directly quantify the mutual information (MI) between sets of parameters and features [23], our method aims to quantify the relationship between model parameters and features for individual observations. Computing estimates of the MI for simulator models is difficult without access to a closed form likelihood and purely sample based estimators [48, 49] do not allow for inspection of individual or small sets of observations. However, this is of interest when different cells, cell types or cell families are constrained by different feature sets. Nonetheless, FSLM still retains the option to obtain a single ranking of features across a whole dataset by averaging over all observations in the set, which would then be equivalent to the MI between posterior and prior (the trained neural network is amortized and, thus, requires no retraining for different observations).

For the case of a Hodgkin-Huxley model, frequently used in neuroscience as a mechanistic model of neural activity, we showed that action potential threshold and action potential count were the most informative features, jointly constraining the action potential threshold, the resting potential, membrane capacitance as well as model conductances (except for that of potassium). Removing these features from the inference procedure led to much more poorly constrained posteriors and simulations that did not match the original trace.

We note some limitations of FSLM and the metrics introduced: First, the inherent dependence on MDNs for the conditional neural density estimation limits overall accuracy for highly complex posteriors or likelihoods, compared to e.g. normalizing flows, as it only works well for likelihoods that are well approximated by mixtures of Gaussians. Additionally, the use of full covariance matrices can be problematic in higher-dimensional feature spaces due to the curse of dimensionality and one would have to evaluate whether the use of e.g. diagonal covariances is more suitable. Second, since our metrics are relative to the overall level of uncertainty in a given posterior marginal, for very sharp distributions a small increase in variance can already lead to a large increase in our metrics, which would not be the case for a broad distribution. While this may be the desired behavior in some cases, it might be misleading in others. Third, the effect of a feature may be judged differently depending on whether the effect of removing it from a large set of features is being assessed, or whether it is being added early in a greedy feature selection run. Finally, the approximation quality of NLE, thus also FSLM, depends crucially on the choice of prior [50]. However, posterior predictive checks [51, 52] and coverage tests [53] applicable to NLE can also be applied to FLSM to verify the correctness of the inference results.

Our method opens the door for careful studies on the relationship between data features and model distributions. In neuroscience, for example, we could explore whether commonly used features such as those used here constrain the posterior equally across different cellular families or types — for

example, is action potential threshold equally important for modeling fast-spiking interneurons and for cellular families with more complex firing behavior, or are additional features needed in this case? Given the efficient exploration of the relationship between features and posterior uncertainty, we believe that our method will also be applicable to a wide range of inference problems beyond neuroscience.

## Acknowledgments

We thank the German Research Foundation for support through the Excellence Cluster "Machine Learning — New Perspectives for Science" (EXC 2064, 390727645) and a Heisenberg Professorship to PB (BE5601/8-1). JHM and PB are also members of the Tübingen AI Center funded by the German Ministry for Science and Education (FKZ 01IS18039A). We thank the International Max Planck Research School for Intelligent Systems (IMPRS-IS) for supporting Michael Deistler. We also thank our colleagues, in particular Jonathan Oesterle and Lisa Koch, for their valuable comments and fruitful discussions.

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
