# A Appendix

## A.1 Linear Gaussian Model

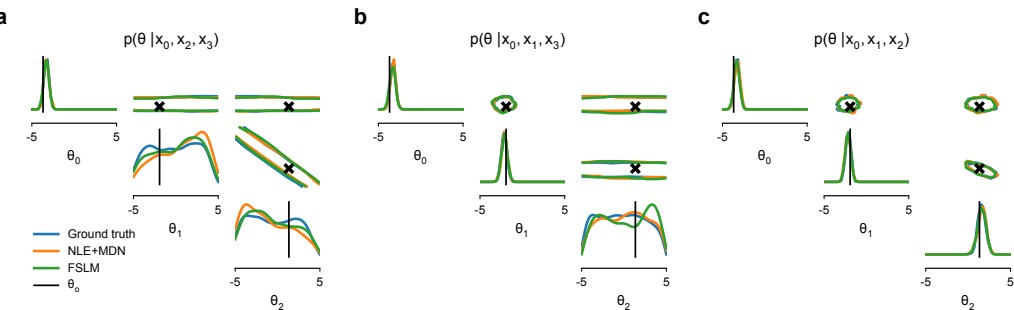

Figure 5: 1D and 2D marginals of the LGM posteriors with one feature marginalized out as estimated by NLE and FSLM compared to the ground truth, which can be analytically computed. **a.-c.** Posterior distributions when $x_1$, $x_2$ and $x_3$ are removed from the feature set, respectively. The (in)dependence of $\theta$ and $x$ of the LGM are reflected in the the 1D marginals assuming the uniform density of the prior when $x_i$ are removed.

## A.2 Excluding invalid simulations

The simulator for the HH model can produce voltage dynamics that do not exhibit any spiking. This will lead to undefined values (NaN) in the spike related summary features that we use (see Tab. 3). In direct posterior estimation methods [15, 16], the invalid training pairs can simply be ignored and dropped from the dataset. However, this is not the case for methods learning the likelihood, as this leads to likelihood estimates that are unaware of parameter regions which produce invalid simulations and hence a biased estimate according to

$$\ell_\phi(\boldsymbol{x}_o|\boldsymbol{\theta}) = \ell_\phi(\boldsymbol{x}_o, y = valid|\boldsymbol{\theta}) \approx p(\boldsymbol{x}_o|\boldsymbol{\theta}, y = valid) = \frac{p(\boldsymbol{x}_o, y = valid|\boldsymbol{\theta})}{p(y = valid|\boldsymbol{\theta})}, \qquad (9)$$

with $\ell_\phi$ being the biased likelihood estimate. To compensate for this mismatch we therefore employ two techniques also employed by Lueckmann et al. [15] and Glöckler, Deistler, and Macke [34] (for proofs see Glöckler, Deistler, and Macke [34] A.6; Theorem 1, Lemma 1 and Lemma 3).

First, we restrict the prior to reduce the fraction of invalid training examples that are produced in the first place, by training a probabilistic classifier to only accept parameter samples that are predicted to result in valid observations. The classifier we use is just a simple fully connected neural net with a softmax output, which is trained with a negative log-likelihood loss. Hence the restricted prior comes out to be

$$\tilde{p}(\boldsymbol{\theta}) = p(\boldsymbol{\theta}|y = valid) = \frac{p(y = valid|\boldsymbol{\theta})p(\boldsymbol{\theta})}{p(y = valid)} \qquad (10)$$

, where $p(y)$ is the probability of $\boldsymbol{\theta}$ producing valid observations and $p(y|\boldsymbol{\theta})$ is the binary probabilistic classifier that predicts whether a set of parameters $\boldsymbol{\theta}$ is likely to produce observations that include non valid (NaN or inf) features. We only expend a small amount of simulation budget to do this.

Since, a small fraction of samples still contains non-sensical observations which have to be withheld from training, we still have to adjust for the remaining bias of the likelihood $\ell_\psi$. This is achieved by estimating $c_\zeta(\boldsymbol{\theta}) = p(y = valid|\boldsymbol{\theta})$ on the entirety of the training data, such that:

$$\ell_\psi(\boldsymbol{x}_o|\boldsymbol{\theta})p(\boldsymbol{\theta})c_\zeta(\boldsymbol{\theta}) \propto p(\boldsymbol{x}_o|\boldsymbol{\theta})p(\boldsymbol{\theta}) \propto p(\boldsymbol{\theta}|\boldsymbol{x}_o) \qquad (11)$$

We implement $c_\zeta(\boldsymbol{\theta})$ as a simple logistic regressor that predicts whether a set of parameters $\boldsymbol{\theta}$ produces valid observations features. Unlike Glöckler, Deistler, and Macke [34] however, we opt to combine this calibration term with the prior, rather than the likelihood. This is mathematically equivalent according to Eq. 11 and is easier to implement in practice.

## A.3 The Hodgkin Huxley Model

The gating dynamics of the HH model are expressed through Eq. 12, 13, where $\alpha_z(V_t)$ and $\beta_z(V_t)$ denote the rate constants for one of the gating variables $z \in \{m, n, h, q, r\}$. They model the different voltage dependencies for each channel type and their parameterisation depends on the specific model that is used. In this work we stick to the kinetics of $\alpha_z(V_t, V_T), \beta_z(V_t, V_T), p_\infty(V_t)$ and $\tau_p(V_t, \tau_{max})$ as modeled in Pospischil et al. [44].

$$\frac{dz_t}{dt} = (\alpha_z(V_t)(1 - z_t) - \beta_z(V_t)z_t)\frac{k_{T_{adj}}}{r_{SS}} \tag{12}$$

$$\frac{dp_t}{dt} = ((p_\infty(V_t) - p_t)/\tau_p(V_t)) k_{T_{adj}} \tag{13}$$

In our experiments we simulate response of the membrane voltage to the injection of a square wave of depolarizing current of $200\ pA$. Since the system of equations is not solvable analytically, we compute the membrane voltage $V_t = V(t)$ iterating over the time axis in steps of $dt = 0.04\ ms$ using the exponential Euler method [54] as it provides a good trade-off between computational complexity and accuracy. The initial voltage is chosen as $V_0 = 70\ mV$ the membrane voltage $V(t)$ is simulated for a time interval of $800\ ms$.

For the parameters we used the following values or prior ranges:

| Parameter | Lower bound | Upper bound |
|---|---|---|
| C | $0.4\ \mu F/cm^2$ | $3\ \mu F/cm^2$ |
| $g_{Na}$ | 0.5 mS | 80 mS |
| $g_K$ | $1 \cdot 10^{-4}$ mS | 30 mS |
| $g_M$ | $-3 \cdot 10^{-5}$ mS | 0.6 mS |
| $g_{leak}$ | $1 \cdot 10^{-4}$ mS | 0.8 mS |
| $g_L$ | $-3 \cdot 10^{-5}$ mS | 0.6 mS |
| $\tau_{max}$ | 50 s | 3000 s |
| $V_T$ | -90 mV | -40 mV |
| $E_{leak}$ | -110 mV | -50 mV |
| $r_{SS}$ | 0.1 | 3 |

| Parameter | Value |
|---|---|
| $E_{Na}$ | 71.1 mV |
| $E_K$ | -101.3 mV |
| $E_{Ca}$ | 131.1 mV |
| $T_1$ | $36°C$ |
| $T_2$ | $34°C$ |
| $Q_{10}$ | 3 |
| $\tau$ | 11.97 ms |
| $R_{in}$ | $126.2\ M\Omega$ |

Table 2: **left**. Prior ranges for the variable parameters of the HH model. **right**. Values for the fixed model parameters.

where $A_{Soma} = \frac{\tau}{CR_{in}}$, C the membrane capacitance, $R_{in}$ the input resistance. Furthermore, $\tau$ is the membrane time constant, $\tau_{max}$ the time constant of the slow $K^+$ current and $V_T$ the threshold voltage. We also include two additional constants, $k_{T_{adj}} = Q_{10}^{\frac{T_2-T_1}{10}}$ and $r_{SS}$, which adjust for the ambient temperature and for different rate scaling respectively.

The output of the HH model is summarized in terms of summary features, which are explained in Tab. 3.

## A.4 The full posterior estimates for the HH model

Here we provide the 1D and 2D marginal plots for all parameters of the posterior estimates $\hat{p}(\boldsymbol{\theta} \mid \boldsymbol{x})$, $\hat{p}(\boldsymbol{\theta} \mid \boldsymbol{x}_{\backslash APT})$ and $\hat{p}(\boldsymbol{\theta} \mid \boldsymbol{x}_{\backslash APA})$ in Fig.3**c,d**.

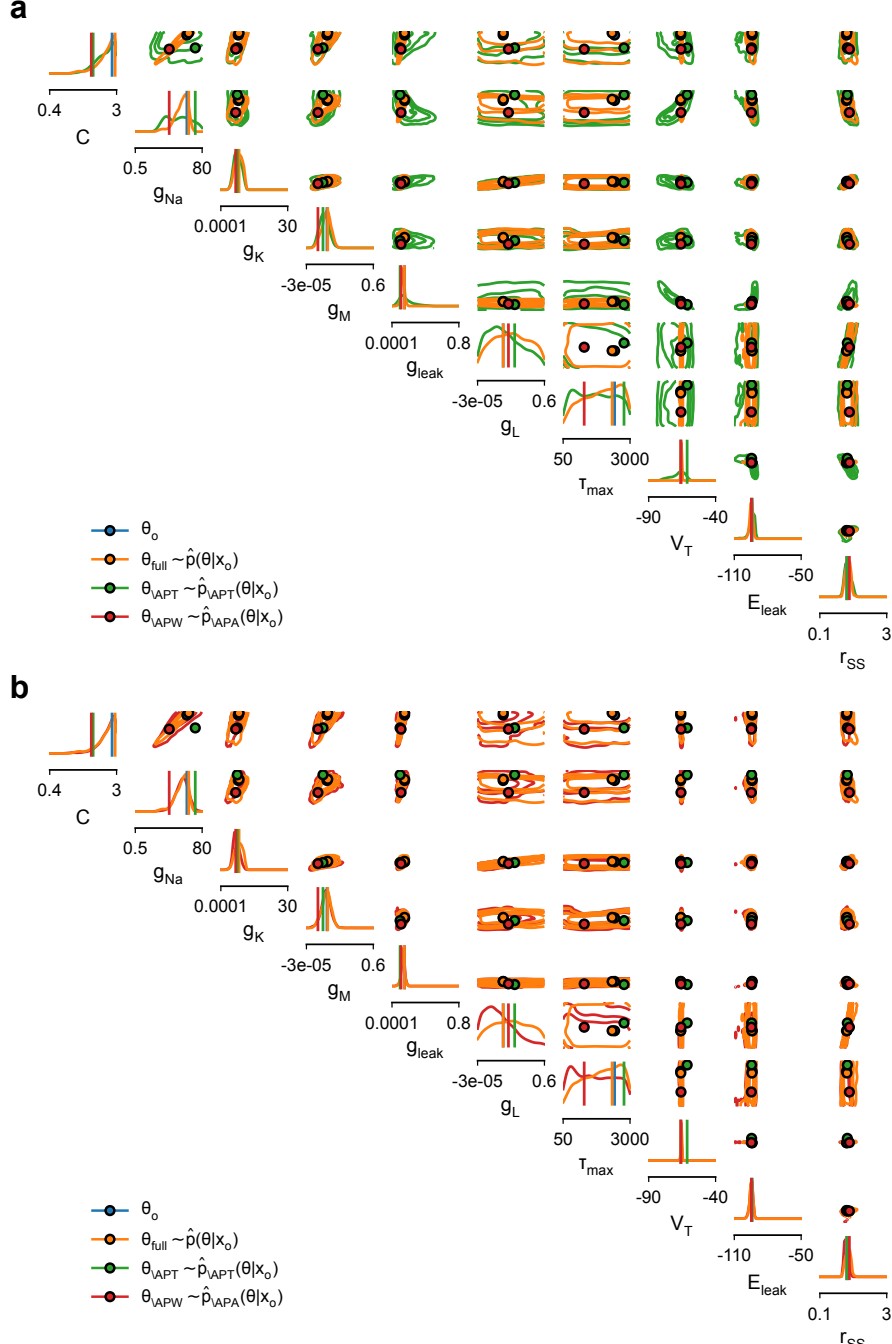

Figure 6: Posterior distributions for all parameters of Fig. 3**c, d**: **a.** The posterior was conditioned on the full feature set (orange) and after the removal of the highly constraining feature APT (green) all of the 1D and 2D marginals are much more uncertain compared to the full posterior (orange). **b.** Shows the same comparison, but after removing the less informative feature APW (red). The posterior distribution without APA (red) is very similarly constraint to the full posterior (orange).

| Feature | Explanation |
|---------|-------------|
| APT | Threshold of $1^{st}$ AP |
| APA | Amplitude of $1^{st}$ AP |
| APW | Width of $1^{st}$ AP |
| AHP | Afterhyperpolarisation of $1^{st}$ AP |
| $APA_{adapt}$ | Adaptation of AP amp. |
| latency | Latency of $1^{st}$ AP |
| APC | Spike count |
| $\mu(V_m)$ | Mean of $V_m$ |
| $Var(V_m)$ | Variance of $V_m$ |
| $\mu(V_{rest})$ | Mean resting $V_m$ |

| Feature | Explanation |
|---------|-------------|
| $APT_3$ | Threshold of $3^{rd}$ AP |
| $APA_3$ | Amplitude of $3^{rd}$ AP |
| $APW_3$ | Width of $3^{rd}$ AP |
| $AHP_3$ | Afterhyperpolarisation of $3^{rd}$ AP |
| $APC(T_{1/8})$ | AP count of first 1/8 |
| $APC(T_{1/4})$ | AP count of first 1/4 |
| $APC(T_{1/2})$ | AP count of first 1/2 |
| $APC(T_{2/2})$ | AP count of second 1/2 |
| $\mu(APA_{adapt})$ | Mean adaptation of AP amp. |
| $CV_{APA}$ | Coeff. of Variation of AP amp. |
| $ISI_{adapt}$ | Inter-spike-interval adaptation |
| $CV_{ISI}$ | CV of ISI |
| $\sigma(V_{rest})$ | Standard deviation of resting $V_m$ |

Table 3: Brief explanation of HH-model summary statistics. AP = action potential; amp = amplitude, $V_m$ = membrane potential. **left**. Statistics used for training of all HH posteriors. **right**. Additional set of statistics, that was only used in the baseline posterior estimate, which we use to compare methods in Tab. 1.