# OpenReview forum: "Efficient identification of informative features in simulation-based inference"
_NeurIPS.cc/2022/Conference — NeurIPS 2022 Accept_

### Official Review · Reviewer_STJL · 2022-07-06

**Rating:** 5
**Confidence:** 5
**Soundness:** 3 good
**Presentation:** 2 fair
**Contribution:** 2 fair

**Summary:**

The paper introduces a method to efficiently estimate the importance of several features / outcome variables of a simulation model in a Bayesian setting. The method is based on mixture density networks and the fact that marginalization of Gaussian distributions is analytically tractable and relatively simple.
The authors use their method to demonstrate in 2 practical examples that the method reliable identifies feature importance and demonstrate the method's superior speed.

**Questions:**

- What kind of density estimator was used for NLE? Just an MDN? Why not a flow? Have alternatives different from the default be tried? This should be mentioned. (Could any of those choices reasonable lead to NLE working better than your method, e.g. a flow instead of an MDN.)
- How is time for training compared? Is there a predefined number of iterations or are you using a stopping criterion? Or is this just to say that you train one MDN and marginalize instead of retraining an MDN on the subset? (I.e. if you used a flow for NLE it's not clear how to compare time.)
- Are you modelling the MDN with a full covariance matrix? If so, what would you suggest for higher dimensional data spaces?


**Limitations:**

The authors discuss some limitations in the discussion section. While there are some elements on the limitations of MDNs, I'd like to know more about the covariance matrix that has to be estimated and hence the scalability. (A reasonable argument to be made would be that scalability isn't that important since handcrafted summary statistics are usually of limited dimensionality, but I think this should be discussed.)

**Strengths And Weaknesses:**

__Strengths__

- Reasoning about the importance of _interpretable_ summary statistics is commonly (and unjustly) ignored in the era of neural networks. The authors therefore tackle an issue that is highly relevant for reliable and trustworthy inference.
- Marginalisation using the MDN is indeed an efficient approach and using the so computed marginal distributions to compute the feature important makes sense to me.
- The method is easy to understand and implement
- The writing is coherent and what is said is easy to follow (but the text does lack crucial information, see below)

__Weaknesses__

- The paper seems rushed: there is very little information on many relevant details. For example, what kind of density estimator was used for NLE? A flow? If so, what kind of flow? How many layers? I have many questions, see below. If any of those are answered in the text and I missed them, could the authors please let me know?
- The paper contains only 2 examples, one of which is just a simple Gaussian. I understand the value of the Gaussian example, but the experiments section then only has one "real" model, which puts this section on the lighter side. (Perhaps the relatively long description of the HH model could have gone to the appendix in favour of another example.)
- The literature review is a bit biased towards more recent developments and perhaps a bit unfair towards earlier contributions. For example:

> likelihood-based Bayesian methods such as variational inference [8] or Markov Chain Monte Carlo [9] cannot be used.

That's a bit imprecise. Of course, MCMC can be used, in the form of ABC-MCMC [1] or pseudo-marginal methods [2]. The argument against those methods is more the computational cost and potentially slow convergence, not however, that it's impossible.

- Equation 11 in the appendix is confusing. The RHS has a variable $valid$ which the LHS does not have. The reference for proofs is to imprecise, I could not find the relevant part. The reference should point to the theorem that establishes the result.

[1] Marjoram, P., Molitor, J., Plagnol, V., & Tavaré, S. (2003). Markov chain Monte Carlo without likelihoods. Proceedings of the National Academy of Sciences, 100(26), 15324-15328.

[2] Andrieu, C., & Roberts, G. O. (2009). The pseudo-marginal approach for efficient Monte Carlo computations. The Annals of Statistics, 37(2), 697-725.

---

> ### Author Response · Authors · 2022-08-02
> **Response to STJL**
>
> We thank the reviewer for their comments and appreciate the valuable feedback. The reviewer found the importance of evaluating “interpretable summary statistics” crucial for trustworthy inference and our method easy to follow. However they also expressed some concerns regarding the use of mixture density networks (MDNs) with respect to high dimensional problems and in comparison to flows as well as pointing out biases towards recent literature, which we discuss and address below.
>
> __Density estimator architecture and flow-based NLE__: The reviewer asked about details of the NLE architecture and training. To show that  FSLM achieves the same posterior accuracy via marginalized likelihood estimates as neural likelihood estimates trained directly on marginal feature data, while being significantly more efficient, we opted to use MDNs for both algorithms. We make this more explicit in the revised version (line 165), expanding on our previous mention in Fig. 1. We also included more detailed information about the number of layers (line 166).
>
> With regards to the applicability of flows, we now show that running the brute-force feature selection with a flow-based NLE is slower than the MDN-based NLE. We added this information to Table 1. Of course, flows are the state-of-the-art for NLE in terms of performance. Performing new analysis, we show that FSLM yields very similar feature importance estimates as the significantly slower brute-force flow-based NLE approach (see Table 1), indicating that FSLM-derived features are consistent with the features identified with flow-based NLE architectures. We will add this information to Fig. 3 and associated text in the final version of the paper (we cannot do so currently, due to space constraints).
>
> __The use of MCMC for likelihood free simulator models__: We thank the reviewer for pointing out our imprecise wording here, we changed the wording to be more specific in the revised version (line 28).
>
> __Comparing training times__: The reviewer is asking for details about how training time was exactly measured. This is indeed an important piece of information, which we have added to the revised submission (line 168). The training times are compared based on the following convergence criterion: We split the data into a training and a validation set with a 90:10 split. We then stop the training when the log likelihood has not improved over 20 epochs on the validation set. In our experiments, this yielded very accurate posterior estimates and ensured sufficient convergence of the density estimators. As correctly pointed out by the reviewer, the neural likelihood estimates obtained on the entire feature set can be reused for FSLM, hence the training times for FSLM are essentially the same, the only difference being that no additional training for further marginal likelihoods is needed. This is also visible in Fig. 1.
>
> __High dimensional covariance matrices__: We used full-rank covariance matrices in this work and did not explore further whether restricted covariance matrices would yield better performance. For higher-dimensional features spaces, one clearly would have to evaluate whether e.g. diagonal covariance matrices are more suitable. We added these possibilities to the Discussion (lines 277 - 279).
>
> __Focus on one experiment__: The reviewer suggested investigating a different model in addition to the HH model. As the HH-model application is of high value to computational neuroscience and inference in this model is quite hard due to the involved non-linearities, we chose to focus on this example and analyze it in depth.
>
> __Missing references__:  We thank the reviewer for the additional references which we added to the paper.
>
> Inconsistent notation and missing cross-references: We thank the reviewer for pointing this out, we have made the notation more consistent in the revised version of our submission and added pointers to the appropriate theorems (line 515).

---

### Official Review · Reviewer_Pf14 · 2022-07-08

**Rating:** 6
**Confidence:** 4
**Soundness:** 3 good
**Presentation:** 4 excellent
**Contribution:** 2 fair

**Summary:**

The authors propose a new method for determining informative features in simulation based inference. Recently proposed simulation based inference methods proceed by (1) generating parameter samples from the prior, (2) simulating to generate conditional samples of the data given the parameter, (3) fitting an approximation to the likelihood of data features under the simulator model consisting of a mixture model parameterized by neural networks and (4) sampling from the posterior over parameters given the data, using the conditional mixture model as an approximation to the true likelihood. The authors observe that to obtain a posterior conditional on a subset of features, one can use the property of multivariate normals that their marginals are analytically tractable, and replace the full approximate likelihood with the marginal approximate likelihood.

**Questions:**

There are some serious typos and lack of clarity in Eqn. 6 - there's min_j, but j does not appear; lowercase m and n are undefined; and it's unclear what min^N_j means (min raised to the Nth power??).

More generally I'm confused by the motivation for the KL estimator. KL estimation from paired samples is notoriously difficult and unreliable in high dimensions. I think it would make more sense to use maximum mean discrepancy or related techniques, which are well-established for moderately high dimensional data, and  which offer the further advantage of providing a witness function that can help determine where the two distributions differ. (see https://jmlr.csail.mit.edu/papers/v13/gretton12a.html and https://papers.nips.cc/paper/2015/hash/0fcbc61acd0479dc77e3cccc0f5ffca7-Abstract.html)





**Limitations:**

The authors discuss some limitations. One additional limitation that I think is important is that the success of NLE in general depends crucially on the choice of prior; if the true parameters are far out in the tails of the prior, the likelihood approximation is likely to be very poor, particularly as it restricts the tails of the likelihood to be Gaussian (and will thus fail if the true likelihood has, for instance, very fat tails). I worry that the proposed method does not take into account uncertainty or systematic errors in the approximation quality

**Strengths And Weaknesses:**

The method itself is very straightforward, enough so that it seems like a trick any researcher in the field would be able to figure out, as it relies on well-known properties of multivariate normal distributions. However, the method is presented very clearly. Identifying informative features in simulation based inference is an important and poorly studied problem (to my knowledge); this paper offers smart and clear problem statement and setup. Moreover, writing a paper about the method helps motivate the particular advantages of neural likelihood estimation in contrast to competing techniques (such as NPE).

The evaluations are clear and compelling, especially the biological application. Hodgkin Huxley models are notoriously hard to fit, and the connection between the features that electrophysiologists focus on and actual underlying ion channel mechanisms are notoriously difficult to understand. This paper thus offers a compelling advance for the computational neuroscience/electrophysiology field in particular.

---

> ### Author Response · Authors · 2022-08-02
> **Response to Pf14**
>
> We thank the reviewer for their comments and appreciate the valuable feedback. The reviewer found our algorithm to be a “compelling advance for the computational neuroscience field”, solving a notoriously hard problem in computational neuroscience. Below, we further justify the choice of KL-divergence as a measure and discuss the importance of the prior.
>
> __The use of KL over MMD__: The reviewer suggests to use the MMD as an alternative to the KL divergence.  We had run preliminary experiments with both MMD- and KL-estimators and found that the KL-estimates lead to a much more consistent ranking of features. Nevertheless, the MMD remains an interesting direction to explore in future work.  We added a short statement regarding our initial results with MMD to the text (see lines 146 - 148).
>
> __Importance of prior__: We thank the reviewer for bringing this up. As this is a limitation inherited from NLE, rather than being specific to FSLM, it can be remedied by employing posterior predictive checks and coverage tests applicable to NLE [1-3], which take into account uncertainty and systematic errors in approximation quality. We have added this as a point in our discussion. (see lines 285 - 288)
>
> [1] Talts S, Betancourt M, Simpson D, Vehtari A and Gelman A Validating Bayesian Inference Algorithms with Simulation-Based Calibration 24
> [2] Gabry J, Simpson D, Vehtari A, Betancourt M and Gelman A 2019 Visualization in Bayesian workflow Journal of the Royal Statistical Society: Series A (Statistics in Society) 182 389–402
> [3] Zhao D, Dalmasso N, Izbicki R and Lee A B Diagnostics for Conditional Density Models and Bayesian Inference Algorithms 11
>
> __Typos and misleading notation in formulas__: We apologize and have revised them in the updated submission.

---

### Official Review · Reviewer_9rkA · 2022-07-11

**Rating:** 6
**Confidence:** 4
**Soundness:** 4 excellent
**Presentation:** 3 good
**Contribution:** 2 fair

**Summary:**

The paper proposed a method to determine the "informativeness" of data features by comparing the width of posteriors with some data features marginalized out. The authors compare a naive solution where many models are trained to their proposed technique which takes advantage of the properties of Mixture Density Networks (DMN) as the architecture for Neural Likelihood Estimation (also known as synthetic likelihood) to trivially marginalize over data features without retraining.

Their method Feature Selection Through Likelihood Marginalization (FSLM):
- allows for closed-form marginalization of data features, thereby enabling access to posteriors which are conditioned on arbitrarily marginalized data random variables without retraining.
- comes with a greedy feature selection strategy to select useful features from a predefined set.
- is compared to ground truth posteriors from simple linear Gaussian model and against a non-linear neuronal model.

**Questions:**

- Why is this method superior to a method which estimates the mutual information between parameters and data with some data features marginalized out?


**Limitations:**

- I think this was also made clear.

**Strengths And Weaknesses:**

## Originality

### Strengths
- The closed-form marginalization is simple and easily seen as a result of using a MDN. I've not seen it used before and I find it interesting for their purposes.

### Weaknesses
- The paper does not reference the work of Benjamin Wandelt and coauthors which aims to transform simulation data into a compressed format which maximizes the information shared by the data and its compressed representation for Neural Likelihood Estimation. This is not strictly the same subject as Wandelt's work does not quantify the value of specific dimensions of input data; however, it is a highly relevant dimension reduction technique. Please consider citing all of the papers on the [PyDelfi](https://pydelfi.readthedocs.io/en/latest/intro.html) website (there are three).
- There is other tangentially related work regarding marginalizing over parameters. In particular, [Truncated Marginal Neural Ratio Estimation](https://arxiv.org/abs/2107.01214), [Arbitrary Marginal Neural Ratio Estimation for Simulation-based Inference](https://arxiv.org/abs/2110.00449), and (from pydelfi) [Nuisance hardened data compression for fast likelihood-free inference](https://arxiv.org/abs/1903.01473). Perhaps it would be interesting to include for the purposes of dimensionality reduction.

## Quality
### Strengths
- The Gaussian linear model example is chosen well to show the relevance of independence between parameters and data and the corresponding effect on the estimated posterior.
- The posterior visualization, tables which indicate computational cost and estimated KL divergence, and search result visualizations are all quite clear.
- The limitations of using MDNs are made clear and the benefit for feature search is as well.

### Weaknesses
- Although the method seems nice, I still don't understand why the practitioner would not instead want to compute the mutual information. We do not know what sort of observations we will have in the future and the mutual information is likely more robust to future data (rather than overfitting feature selection to x_o.)

## Clarity
- Clarity was satisfactory in my opinion.


## Significance
### Strengths
- They're certainly a nice feature of MDNs and NLE.

### Weaknesses
- MDNs seem to be of limited value in the times of normalizing flows, etc.
- I also don't see why setting the feature importance based off a single x_o is really so interesting. Selected features should have some sort of predictive power for future data, right? If so, shouldn't we then integrate over all possible data, i.e. estimate the mutual information between different feature sets and the parameters?

---

> ### Author Response · Authors · 2022-08-02
> **Response to 9rkA**
>
> We thank the reviewer for their comments and appreciate the valuable feedback. The reviewer found our approach to be interesting and novel, but suggested mutual information-based strategies for feature selection, which we now discuss in more detail. Also, they asked for the generalization to flow-based NLE, for which we performed new analyses and show that the identified features generalize well.
>
> __Estimating mutual information between parameters and features__: The reviewer suggested that an alternative to our proposed algorithm would be to estimate the mutual information (MI) between sets of features and parameters for the model. We agree that this would be an interesting strategy, and we have added a paragraph in the Discussion to address this. FSML and MI are closely related. Using the definition of the MI
>
> $MI = D(p(t, x) || p(t) p(x)) = \int p(x) \int p(t | x) log(p(t|x)/p(t)) dt dx = E_x[D(p(t|x) || p(t)]$
>
> reveals that the mutual information is the average D_KL between the posterior and the prior. Thus, the MI can be recovered by averaging FSML over the entire dataset and using the D_KL between posterior and prior as distance metric. We note that FSML could easily be extended to the entire dataset by averaging over many observations (the trained neural network is amortized and, thus, requires no retraining for different observations).
>
> In many real-world scenarios such as computational neuroscience, one is particularly interested in the importance of features for individual observations. This is because different features may be important in different cell types or cell families. We, therefore, opted to employ FSML as a method to identify important features of individual observations, but as stated above, our method could easily be extended to average over the entire dataset (and, in that case, corresponds to estimating the MI). We emphasize that, in general, estimating MI in simulation-based models is difficult because these models have no closed-form likelihood and, thus, would have to rely on sample-based approximations to the mutual information. We have highlighted the similarities and differences between MI and FSML in the discussion and hope that this allows users to pick the ideal method for the problem at hand.
>
> __The choice of MDNs over Flows__: We agree with the reviewer that flows have advantages with regards to performance when compared to MDNs, as we discuss in our paper. However, we also note that “even in times of normalizing flows” they still remain a relevant tool in similar settings, see e.g. [1] used MDNs for inference on HH models. Nonetheless, we have now  performed an additional new analysis of how well our MDN-derived features generalize to a NLE with flows. To this end, we estimated feature importance in the brute-force setting using a flow-based NLE. We found that MDN-derived features generalize well, with high agreement between the two architectures and only one  additional feature (mean resting potential) showing up in the flow-based NLE. We will add this information to Fig. 3 and the associated text and to Table 1 in the final version of the paper (we cannot do so currently due to space constraints) as well as discuss the limits of this correspondence, the details of which will have to be investigated in future work.  Together, this argues for the usefulness of our method also for flow-based models, exploiting the fact that MDNs allow for efficient and exact computation of the marginal likelihoods.
>
> [1] Gonçalves P J, Lueckmann J-M, Deistler M, Nonnenmacher M, Öcal K, Bassetto G, Chintaluri C, Podlaski W F, Haddad S A, Vogels T P, Greenberg D S and Macke J H 2020 Training deep neural density estimators to identify mechanistic models of neural dynamics eLife 9 e56261, https://elifesciences.org/articles/56261.pdf
>
> __Additional references__: We thank the reviewer for pointing us to the additional references, some of which we have now included in the revised version of our paper. (see line 40)

---

> > ### Comment · Reviewer_9rkA · 2022-08-07
> > **fair reply**
> >
> > Thanks for your reply. I think it all makes sense and I'm glad you added the discussion of MI and how a single observation can be important in neuroscience. I will raise my score.

---

> > > ### Author Response · Authors · 2022-08-09
> > > **response**
> > >
> > > We thank the reviewer for their reply and for appreciating our efforts.

---

### Official Review · Reviewer_GQHb · 2022-07-11

**Rating:** 6
**Confidence:** 3
**Soundness:** 4 excellent
**Presentation:** 3 good
**Contribution:** 2 fair

**Summary:**

This paper develops an efficient feature selection method in simulation-based inference, where each evaluation of the likelihood is costly. Conventionally, to test the sensitivities of the posterior distribution to specific features, one would need to re-evaluate the posterior with different combinations of features many times. The key idea of the method is to estimate the likelihood as an intermediate quantity instead of directly estimating the posterior, and analytically marginalize the joint likelihood for each feature instead of re-evaluating from scratch each time. The marginalization is made possible by the use of mixed density networks with gaussian properties where different parameters are easily partitioned. Using a Hodgkin-Huxley model as an example, the paper demonstrates how this method can be used for selecting features at a much smaller computational cost.

**Questions:**

i) It would be nice to have a better "chance level" curve for the random selection in Fig 4a, i.e., average over multiple random sequences of features.

ii) The speedup by the proposed method is evident, but 20 hours is still a long time if one wants to repeat. Can the authors comment on how/where the algorithm can use parallelization to reduce computation time?

iii) I had two questions, and they both boil down to understanding the computation time cost of considering more features:

- There is still a human curated step required in the procedure, namely the pre-selected feature list. Can the authors comment on how this new level of efficiency could change the constraints related to the choice of the pre-selected feature list? That is, can one now run through a very long list of candidate features, including all the silly things that one would not expect to be important a priori?
- Regarding the HH model experiment: I agree that the greedy features selection is a completely reasonable thing to do. But looking at Fig 4a made me wonder if This method also provides easy ways to explore combinations of features. (Specifically, from Fig 4a, I was wondering if the combination of APT, AHP, and APW were important, based on the drop in the blue curve at the addition of APT, and the drop in the black curve at the addition of APW.) How much additional cost would it be to investigate combinations of features?

**Limitations:**

The authors have adequately addressed the limitations of their methods.

**Strengths And Weaknesses:**

This work seems to be a small but useful technical improvement to simulation-based inference (SBI). I am not an expert of the SBI method space in particular, so I cannot speak much on the originality or the novelty of the idea. But I have seen usages of SBI and surrogate likelihood/posterior approaches in neuroscience as well as in other domains (computational chemistry), and I have not seen an analytical marginalization approach like this before. Whenever the mixed gaussian model (with the convenient partitioning properties that allow marginalization) is a good fit to the system, this method will provide an immediate practical advantage. The paper is mostly clear and well written.

---

> ### Author Response · Authors · 2022-08-02
> **Response to GQHb**
>
> We thank the reviewer for their comments and appreciate the feedback. The reviewer found our work a ‘useful technical improvement to simulation-based inference’. They suggested providing a more informative baseline for our greedy feature search, which we added to Fig. 4a. This additional analysis clearly shows that our greedy feature selection is more efficient than the baseline. Furthermore, we reply to a few questions regarding the computational cost of our algorithm - here, we emphasize that the speed-up achieved enables much more efficient search for good features for simulation-based inference.
>
> __Better “chance level” for greedy search__: We thank the reviewer for this suggestion. We agree that multiple random sequences are much more informative and we included this in the revised version of Fig. 4a. The revised figure clearly shows the advantage of our method in selecting features to obtain an accurate posterior.
>
> __Parallelization of the algorithm__: Regarding the computational efficiency of the algorithm, we would like to stress that our algorithm achieves a 12x speedup in training time and a 5x speedup in total time (including sampling) over the naive implementation. As the reviewer noted, these are quite significant. Further speed-ups could be achieved by running the sampling for all subsets of features in parallel. This would reduce the total time (training + sampling) for FSLM from $20.48 \pm 2.08$ h to$ 8.42 \pm 1.67$ h. Furthermore, the sampling process itself could be spread out across several threads, by running multiple MCMC chains in parallel.
>
> __Constraint on the number of features__: The reviewer asks whether our algorithm enables exploration of a very long list of features to minimize human intervention. Indeed, our algorithm makes that possible. The longer the list of features, the greater the advantage of our method from a computational standpoint, compared to repeated runs of NLE. In that case, however, it would be advisable to use the greedy feature search set-up. Here, more features only expand the search space, but the initial improvement in KL for the first few features will be large, so selection can be robustly done. In contrast, for the setup of testing the effect of removing individual features from a large feature set, the differences in KL estimates and the IQR ratios of the marginals become increasingly small such that detecting small changes in uncertainty can become difficult.
>
> __Feature combination for HH model__: The reviewer asks for the computational cost of evaluating feature combinations. Indeed, there is no additional cost in investigating combinations of features, since the likelihood can be marginalized analytically with regard to one, two or more features. This also implies that instead of only expanding the most promising node in the search tree, as was done in our greedy implementation, the scheme is easily extensible to a full beam search, where the N-most promising nodes are expanded at each iteration. This would allow us to keep track of potentially combined effects of different features. We added a remark regarding this point in the respective section (lines 233 - 235).

---

> > ### Comment · Reviewer_GQHb · 2022-08-06
> > **response to authors**
> >
> > I would like to thank the authors for the rebuttal comment as well as the revised version of the paper.
> >
> > It is great to see the revised version of Fig 4a with a better-constructed null model (distribution over multiple random sequences).
> >
> > The additional explanations about the computational cost and efficiency of the algorithm make sense, and help me understand the method better.
> >
> > My ratings for the paper remains unchanged; I still think that this is a small but nice technical improvement with clear practical advantage.

---

> > > ### Author Response · Authors · 2022-08-09
> > > **response**
> > >
> > > We thank the reviewer for their reply and for appreciating our efforts.

---

### Author Response · Authors · 2022-08-02
**Summary of response**

We thank all reviewers for their comments and appreciate the feedback, which helped us improve the manuscript. They found our work interesting, novel and clearly written, providing a “useful technical improvement to simulation-based inference” and a “compelling advance for the computational neuroscience field”. In particular, they highlighted the importance of evaluating “interpretable summary statistics” as crucial for trustworthy inference. We thank the reviewers for this very positive assessment.

Of course, the reviewers also voiced criticism, which we will in detail address in the point-by-point replies below. We would like to highlight two key issues, for which we performed new analyses prompted by the reviews. First, one reviewer criticized our choice of baseline for the greedy feature selection pipeline from Fig. 4. We added a new baseline to this figure, averaging over multiple random runs, which now clearly shows that our method is able to select informative features much more efficiently than random search. Second, two reviewers suggested that MDNs are not state-of-the-art for NLE since flow-based NLE typically performs better. We address this point with a new analysis which shows that flow-based NLE is slower than MDN-based NLE when used in the brute-force setting and that the features identified by our method generalize well to flow-based NLE.

Thus, we believe we have addressed the criticism of the reviewers. We would like to underscore the need for an efficient feature selection method for neural-likelihood-based simulation-based inference, which yields interpretable features for simulation-heavy fields such as computational neuroscience. Such a method is provided by FSLM, and we hope our arguments convince the reviewer’s of this fact.

---

### Meta-Review · Area_Chair_2XwJ · 2022-08-23

**Recommendation:** Accept
**Confidence:** Certain

**Metareview:**

The paper presents a method for feature selection in simulation-based inference, that is, for quantifying to what extent each feature (or summary statistic) contributes to reducing posterior uncertainty. As this can be accomplished naively by re-estimating the posterior after systematically omitting features, the focus is on developing a fast method instead. The proposed method is evaluated on parameter estimation of a Hodgkin-Huxley neuron model from neuroscience.

The reviewers found the paper to be clearly written and did not voice major concerns regarding its technical quality. The proposed method is simple, efficient and clearly presented, and the evaluation on the Hodgkin-Huxley model is convincing. A potential drawback of the method is that it requires a model that can be analytically marginalized over, and thus may not benefit from current and future advances in generative modelling.

Seeing as the paper is of good quality without major problems, I'm happy to recommend acceptance.

**Award:**

No

---

### Decision · Program_Chairs · 2022-09-14

Accept